# Can Near-Infrared Spectroscopy Replace a Panel of Tasters in Sensory Analysis of Dry-Cured Bísaro Loin?

**DOI:** 10.3390/foods12234335

**Published:** 2023-12-01

**Authors:** Lia Vasconcelos, Luís G. Dias, Ana Leite, Iasmin Ferreira, Etelvina Pereira, Evandro Bona, Javier Mateo, Sandra Rodrigues, Alfredo Teixeira

**Affiliations:** 1Mountain Research Center (CIMO), Polytechnic Institute of Bragança, Campus de Santa Apolónia, 5300-253 Bragança, Portugal; lia.vasconcelos@ipb.pt (L.V.); ldias@ipb.pt (L.G.D.); anaisabel.leite@ipb.pt (A.L.); iasmin@ipb.pt (I.F.); etelvina@ipb.pt (E.P.); srodrigues@ipb.pt (S.R.); 2Laboratory for Sustainability and Technology in Mountain Regions, Polytechnic Institute of Bragança, Campus de Santa Apolónia, 5300-253 Bragança, Portugal; 3Department of Food Hygiene and Technology, University of Veterinary Medicine, Campus Vegazana S/N, 24007 León, Spain; jmato@unileon.es; 4Post-Graduation Program of Food Technology (PPGTA), Federal University of Technology Paraná (UTFPR), Paraná 80230-901, Brazil; ebona@utfpr.edu.br; 5Post-Graduation Program of Chemistry (PPGQ), Federal University of Technology Paraná (UTFPR), Paraná 80230-901, Brazil; 6School of Agriculture, Polytechnic Institute of Bragança, Campus de Santa Apolónia, 5300-253 Bragança, Portugal

**Keywords:** consumers, meat products, Bísaro breed, food quality assessment, NIR analysis, non-linear SVR models

## Abstract

This study involved a comprehensive examination of sensory attributes in dry-cured Bísaro loins, including odor, androsterone, scatol, lean color, fat color, hardness, juiciness, chewiness, flavor intensity and flavor persistence. An analysis of 40 samples revealed a wide variation in these attributes, ensuring a robust margin for multivariate calibration purposes. The respective near-infrared (NIR) spectra unveiled distinct peaks associated with significant components, such as proteins, lipids and water. Support vector regression (SVR) models were methodically calibrated for all sensory attributes, with optimal results using multiplicative scattering correction pre-treatment, MinMax normalization and the radial base kernel (non-linear SVR model). This process involved partitioning the data into calibration (67%) and prediction (33%) subsets using the SPXY algorithm. The model parameters were optimized via a hybrid algorithm based on particle swarm optimization (PSO) to effectively minimize the root-mean-square error (RMSECV) derived from five-fold cross-validation and ensure the attainment of optimal model performance and predictive accuracy. The predictive models exhibited acceptable results, characterized by R-squared values close to 1 (0.9616–0.9955) and low RMSE values (0.0400–0.1031). The prediction set’s relative standard deviation (RSD) remained under 5%. Comparisons with prior research revealed significant improvements in prediction accuracy, particularly when considering attributes like pig meat aroma, hardness, fat color and flavor intensity. This research underscores the potential of advanced analytical techniques to improve the precision of sensory evaluations in food quality assessment. Such advancements have the potential to benefit both the research community and the meat industry by closely aligning their practices with consumer preferences and expectations.

## 1. Introduction

In this modern digital age, consumers are more active in the search for specific information about the nutritional value and sensory properties of food. Several scientific research works aim to provide outcomes related to quality control (QC), technological processing, traceability and authenticity in food products. In the case of meat products and their derivatives, exceptions are not applied. Pork is one of the most traditionally consumed meats in the world and is known for its quality attributes. In Portugal, there exist three native pig breeds: the Bísaro, the Malhado de Alcobaça and the Alentejano [1]. Bísaro represents a breed of autochthonous Portuguese pigs with Celtic origins and a part of Portugal’s biological, economic and cultural heritage [2]. It is typically produced in a semi-extensive system, with its dietary management relying on locally available agricultural resources [3]. In addition, the Bísaro breed is known for the quality of the meat and fat from these animals, used for the manufacture of various products of excellence and specific qualities that hold designations, such as the Protected Geographical Indication (PGI) and Protected Designation of Origin (PDO). The products that currently enjoy the PGI designation are Salpicão de Vinhais, Chouriça de carne de Vinhais, Alheira de Vinhais, Butelo de Vinhais, Chouriça Doce de Vinhais, Chouriço Azedo de Vinhais and Presunto Bísaro de Vinhais. The product with the PDO is the Carne de Bísaro Transmontano [4]. Dry-cured Bísaro loin currently lacks recognition under any of these quality labels; however, due to its characteristics, it has the potential to also be a product in this list of products. 

Despite the fact that several research works have been carried out in the reformulation of meat products, the traditional processes used to manufacture these meat products are still of the utmost importance. Most of them undergo traditional meat preservation methods, such as salting, drying or smoking. Some examples are typical Mediterranean dry-cured meat processing characterized by dry salting, no smoking and a long drying process, and brine salting and smoking are used in continental parts of Europe [5]. The specific regional conditions for the application of these methods, together with the typicality of the raw material (genetic type, feed, rearing system, etc.), make it possible to obtain particularly diverse dry-cured products. The overall acceptance of dry-cured products, mainly determined according to flavor, taste and odor compounds, depends on a large reaction that results from traditional production methods [6]. In dry-cured products, key attributes are markedly affected by the ripening process and complex chemical and biochemical changes in the main components of raw meat, which contribute to their characteristic aroma and flavor [7]. 

On the other hand, meat products resulting from pork also depend on factors that are intrinsic and extrinsic to the animal. The intrinsic factors include age, weight at slaughter, sex, genetics and physiological state. The factors extrinsic to the animal involve the housing system, feeding techniques, handling, sanitary and environmental conditions, transportation, pre-slaughter techniques, slaughter, post mortem and meat processing [8]. However, the quality of meat products from Iberian pigs strongly depends upon the breed and rearing system, which include a number of different lines, causing great heterogeneity within the same breed. 

Indeed, the information provided to consumers on the impact on health or any other quality aspect can significantly influence their acceptance of meat products [9,10]. Moreover, consumer expectations regarding the perception of safety associated with processed meat, animal welfare, processing techniques and the preservation of traditional production methods [11] denote the importance of utilizing an autochthonous breed, such as the Bísaro, in order to add value to the product. These characteristics, in the classic way, are usually determined via mechanical, physico-chemical measurements and sensory analysis, most of which are invasive, expensive and time-consuming [12]. So, it is essential to conduct rapid assessments of all the meats’ and meat products’ quality traits through physical and sensory tests to ascertain whether the product aligns with the final consumer demands [13,14]. Consequently, the sensory evaluation (SE) of the meat and meat products is a necessary tool to check acceptability by consumers as well as to characterize each product. By incorporating this evaluation, companies can understand consumer preferences, optimize product development and ensure customer satisfaction [15]. It also aids in identifying product defects, evaluating changes in formulation or processing and maintaining consistency in product quality. Overall, SE helps in making informed decisions, enhancing product appeal and achieving success in the market. 

However, there are inherent subjective issues (including assessor selection, attribute generation and context effects, among others) [15] that generate greater variability associated with sensory evaluations due to their reliance on humans’ resources. Human perceptions and interpretations of sensory attributes introduce subjectivity and variability into the process [16]. However, this subjectivity and variability can have less impact when trained tasters and specific tests are applied to evaluate sensory properties. Although this SE is carried out by a panel of trained tasters, it is crucial to carefully select and train sensory marks, use analytic tests (discrimination and descriptive) and establish clear evaluation protocols, and awareness of potential sources of variability are necessary to improve the reliability of sensory evaluations [16]. Descriptive analysis (DA) has been used to characterize and discriminate meat and meat products and is the most reliable sensory method for the sensory characterization of these meat products. Although DA provides accurate and reliable results, the economic effort and time needed to train evaluators is a major disadvantage [15].

With all these in mind, the use of alternative techniques, such as computed tomography (CT) [17], magnetic resonance imaging (MRI) [12], hyperspectral imaging (HIS) or near infrared (NIR) spectroscopy [18,19], must be considered in order to support and complement conventional sensory techniques more quickly. Alternative techniques are increasingly gaining priority due to prompt, easy-to-use and minimal pre-processing requirements, making them suitable for rapid implementation in industrial meat applications [12]. Most studies reveal the NIR spectra potential to provide real-time QC based on distinguishing quality labels; predicting the physico-chemical composition, technological parameters and sensory attributes; and classifying and identifying specific meat and meat products [16,20,21,22,23,24,25,26,27,28].

However, NIR needs discriminating methods to predict the sensory attributes of meat. This is mainly due to factors like the inherent heterogeneity of meat and the complexity of NIR spectra, which demands an understanding of chemometric tools to establish any correlative relation between the generated spectra and the peculiarities of the studied samples [29]. In this sense, pre-treatment techniques such as multiplicative scatter correction (MSC), standard normal variate (SNV), smoothing (SMT), baseline removal and first (1st d) and second (2nd d) derivatives are used to reduce and correct possible interferences related to scattering, baseline shifts, path length variations and overlapping spectral bands. Additionally, multivariate statistical techniques such as principal component analysis (PCA), partial least squares (PLS), the sample projections algorithm (SPA), uninformative variable elimination (UVE), genetic algorithms (GA), the K-nearest neighbors algorithm (KNN), multiple linear regression (MLR), principal component regression (PCR), partial least squares regression (PLSR), support vector machine (SVM) and artificial neural networks (ANN) are applied to simplify modeling purposes and are used for quantitative and qualitative purposes [11,30]. The results obtained are valuable for the companies and researchers in the sense that they are more objective, giving an idea of the quality of the product throughout the real-time analyses. This allows for improvements to be made in order to achieve the desired level of QC, without forgetting that organoleptic properties such as odor, flavor and texture must be considered.

In this framework, the purpose of this piece of research is to evaluate the potential of NIR as a rapid predictor of physiochemical attributes in dry-cured Bísaro loin samples. By collecting the NIR spectra of dry-cured Bísaro loins and using different spectral mathematical pre-treatments and chemometric modeling, the aim is to combine these data with the sensory attributes of sensory evaluation (SE) information. This approach can lead to a new analytical method for the sensory evaluation of dry-cured products, allowing product characterization and helping develop specific breed market products.

## 2. Materials and Methods

### 2.1. Animal Management

This study is part of a project (BISOLIVE) between a research center (Carcass and Meat Quality Laboratory at the School of Agriculture of the Polytechnic Institute of Bragança (LTQCC-IPB), Bragança, Portugal) and a meat manufacturing industry (Bísaro industry—Salsicharia Tradicional, Lda^®^, Gimonde, Portugal) to enhance the value of livestock.

A total of 40 castrated Bísaro breed animals (21 females and 19 males) were randomly selected from a meat manufacturing industry (Bísaro Salsicharia Tradicional, Lda^®^, Portugal) and reared on the farms of Covas and Castro Vicente (Portugal) as part of an extensive production system. The animals were fed a typical commercial diet for autochthonous breeds until they reached an average age of 12 months and a live body weight of 135 ± 4.5 kg (last fattening phase). The animals were stunned, slaughtered and exsanguinated at a local slaughterhouse (Municipal Slaughterhouse of Bragança, Bragança, Portugal) with an average carcass weight of 100 ± 4.5 kg. The procedure was described by Àlvarez-Rodríguez and Teixeira [31] and followed the welfare regulations of EU Council Regulation (EC) No. 1099/2009 [32]. At 24 h post mortem, the carcasses were carefully halved, and the left side was weighed and recorded. They were carried to the LTQCC-IPB for carcass evaluation and meat analysis. Forty parts of *Longissimus thoracis et lumborum* (LTL) muscle samples were obtained between the 7th and 12th rib of the animals. After being collected, the animals’ muscles were refrigerated in a chamber between 2 and 5 °C.

### 2.2. Sample Manufacturing

The traditional drying process in the region was carried out at the Bísaro industry—Salsicharia Tradicional, Lda^®^, Portugal. All dry-cured loins were manufactured on the same day using the same series of steps applied in the typical production of the final product. The first step was the dry salting and seasoning phase with the following ingredients: 1.5% salt, 0.5% paprika, 0.5% garlic and 0.1% oregano. The mixing process took place in a rotating drum for 30 min. After this step, the pieces were placed in a refrigeration chamber between 2 and 4 °C for 4 days. The next phase involved stuffing into collagen casings and drying. The most important step of the drying process involved temperature and relative humidity as key control points. Both changed as the drying time progressed: On the first 15 days, the cuts were subjected to a temperature between 4 and 8 °C with a relative humidity between 80 and 90%. During the next 15 days, the product was subjected to a temperature between 8 and 12 °C and a relative humidity between 70 and 80%. During the last 30 days, the product was subjected to a temperature between 12 and 18 °C and a relative humidity between 60 and 70%. Thus, by increasing the temperature and decreasing the relative humidity, the total drying time was reached (60 days), as reported by Leite et al. [26].

### 2.3. Sensory Analysis

To evaluate the dry-cured loins’ quality, a trained taste panel of nine members who had been previously recruited, selected and trained for DA according to NP (NP-ISO-8586-1, 2001) was employed [33]. Training consisted of two phases: The first involved individual evaluations of dry-cured loin samples from different pork animals, and the second focused on adapting the panel elements to DA in scales and sensory descriptors (muscle color, flavor persistency, flavor intensity, bitterness, acidity, sweetness, saltiness, chewiness, juiciness, hardness, scatol odor, androsterone odor, cured odor, rancid odor, odor intensity, fat distribution, muscle/fat and fat color). A structured scoring scale was used, in which 1 indicates the absence of an attribute and 9 indicates a high intensity of the attribute. The whole process (approximately 8 sessions lasting 1–2 h each) was conducted in a specific tasting room in the Sensory Analysis Laboratory at the Polytechnic Institute of Bragança. The choice, definition and consensus of the evaluation methodology of the descriptors were established during 4 sessions. Standard guidelines (ISO-8589, 2007) [34] were followed, maintaining the room temperature at approximately 20 °C and relative humidity at around 50%. The room was illuminated with white light, and each booth had a white light equally. For each dry-cured loin, 1.5 mm thick slices were cut with an industrial meat slicer machine. All samples were randomly coded with three-digit numbers and were offered in disposable plastic dishes to each taster. Mineral water and unsalted toasted bread were used to clean their palates and remove residual flavors. In order to minimize fatigue, a total of four sessions were conducted. Each session included two samples of dry-cured loins for each taster, resulting in ten different samples per session. Tasters were instructed to carefully observe, smell and taste the samples, paying attention to various sensory aspects. They were then asked to provide judgments on the appearance (color intensity and brightness), odor (intensity and identification), oral texture (hardness, juiciness and chewiness) and flavor (basic taste, flavor intensity, identification and persistence). A 9-point scale with the extremes representing either the minimum (low-intensity sensation) or the maximum (high-intensity sensation) was used for the quantitative attributes for a quantitative descriptive analysis. Odor androsterone and scatol identification ranged from 1 (none) to 9 (higher), general color ranged from 1 (light) to 9 (dark), fat color ranged from 1 (white) to 9 (yellow), hardness ranged from 1 (tender) to 9 (hard), juiciness ranged from 1 (dry) to 9 (moist), and chewability ranged from 1 (easy) to 9 (difficult). The tasters also identified basic taste and flavor persistence from a list of possibilities. The methodology that we used is that described by the standard guidelines (ISO-6658, 2005) [35].

### 2.4. Sample Set and NIRS Analysis

A total of 40 samples of dry-cured Bísaro loins were taken for analysis. The samples were minced and placed in petri dishes with a diameter of approximately 9 cm. For spectral analysis, an FT-NIR Master^TM^ N500 (BÜCHI, Labortechnik AG, Postfach, Flawil, Switzerland) was utilized. The instrument operates within a spectral range of 4000 to 10,000 cm^−1^ with a resolution of 4 cm^−1^ and a 360° rotation system. NIRCal BÜCHI software, version 5.5, was employed to save the spectral data into an Excel^TM^ file. Three spectra were measured for each sample, and these spectra were used to develop the calibration equations.

### 2.5. Data Analysis

The raw spectra were smoothed with a cubic smoothing spline (smoothing parameter 0.01) to remove instrumental noise. Furthermore, as reflectance spectra may carry some variability caused by scattering effects [36], the spectra were pre-treated with four different methods to remove this undesirable effect: multiplicative scatter correction (MSC), standard normal variate (SNV), first derivative (1st d) and second derivative (2nd d). The first and second derivatives were determined using the central finite difference method. These four signal pretreatment methods are conventional procedures that can be tested in combination with various signal normalization techniques to further enhance the optimization of NIR calibration.

The samples’ spectra were partitioned into calibration (67%) and prediction (33%) subsets by applying the SPXY algorithm. SPXY modifies the classic Kennard–Stone algorithm to select samples according to their differences in X (instrumental responses) and Y (predicted parameter) spaces [37].

The regression models used to predict the sensory attributes of the dry-cured loins were obtained using support vector regression (SVR) [38]. Different kinds of SVR were tested (Table 1), and the parameters of the models were defined through a hybrid algorithm based on particle swarm optimization (PSO) combined with pattern search [39]. The objective function of PSO was to minimize the root-mean-square error (RMSECV) obtained via 5-fold cross-validation. Several models using all possible combinations in Table 1 (pre-treatment, normalization, SVR type and kernel) and the PSO-optimized parameters were tested for each sensory attribute.

The partial least squares (PLS) models [40] were also tested to check if a linear model could predict the sensory attributes using the NIR spectra and to compare the performance with SVR models.

All data analyses were carried out in MATLAB R2022b using homemade routines developed by the authors and functions available in the software. LIBSVM 3.3 [41] for MATLAB was employed for SVR.

**Table 1 foods-12-04335-t001:** Tested parameters for SVR models to predict the sensory attributes of dry-cured Bísaro loins.

Pre-Treatment	Normalization *	SVR Type **	Kernel **	PSO Parameters **
MSC	Mean center	ε-SVR	Linear	C
SNV	Autoscale	ν-SVR	Polynomial	ε (for ε-SVR)
1st d	Pareto		Radial Base	ν (ν-SVR)
2nd d	Poison		Sigmoid	γ (except for linear kernel)
	MinMax [−1 + 1]			Intercept (for polynomial and sigmoid kernel)
				Degree (2 to 5 for polynomial kernel)

** Chang and Lin [41]; * Van den Berg et al. [42]; SVR—support vector regression; 1st d—first derivative; 2nd d—second derivative; SNV—standard normal variate; MSC—multiplicative scatter correction; PSO—particle swarm optimization.

## 3. Results and Discussion

### 3.1. Sensory Analysis

#### 3.1.1. Sensory Attributes

For decades, DA has been used for the characterization and discrimination of meat and meat products [14]. For that, the descriptors chosen are vital in these analyses because all specific attributes allow a complete characterization of the dry-cured product under study and describe it in more detail. Also, it was applied with a nine-point hedonic scale (used for the assessment of the linking of the product) because this hedonic test could provide information about different products, textures, compositions, etc., which could help us to better understand the tasters’ answers [15].

Therefore, in dry-cured meat products, the main sensory descriptors are basically grouped by visual appearance, texture and flavor. Among all appearance characteristics, the visual meat color (fat and muscle) was the major criterion used to choose meat appearance [43]. In relation to the texture attribute, it is one of the most important eating quality attributes, and it can be defined as the overall impression related to texture properties. The SE of meat and meat products includes juiciness (defined as the sensation of the water content of the sample and the saliva released with the stimulation of the salivary glands), hardness and chewiness (attributes based on the interaction between the textural characteristics and “mouthfeel”) [44]. Also, the flavor attribute is often perceived when the combination of odor (prior to consumption and retronasal olfaction) and taste perception, along with trigeminal perception, leads to flavor perception [43,44]. In addition, two odors that are likely to be detected in animals with a high live carcass weight and/or that are poorly castrated were analyzed: androsterone and scatol hormones of animal origin [45].

#### 3.1.2. Sensory Data

In this sense, the determined descriptive values of dry-cured Bísaro loins are summarized in Table 2. The mean values, together with the minimums, maximums and standard deviations for the 40 samples analyzed, show that the range of variation was wide enough to guarantee an adequate margin for calibration purposes.

In a general way, it can be concluded that the data obtained from the evaluation of trained tasters give quantitative and objective information despite the wide range. In fact, with the AD test (hedonic evaluation) that was used, it was possible to obtain information on the magnitude of the liking or disliking of these studied sensory attributes.

Several publications describe multiple attributes evaluated during the sensory analysis of meat products [46]. So, regarding texture parameters, hardness is defined as the force required to compress food between molars to achieve deformation [46]. In this way, it is possible to correlate hardness to the proteins in the connective tissue component as well as the myofibrillar component of the meat. Chewiness determines the number of chews required for the meat to be ready for swallowing, and juiciness is dependent on the protein structures/compositions of the muscle fibers and connective tissues and is correlated to the sensation of moisture observed in the initial chewing movements [44,47,48]. It was possible to verify that the intramuscular composition and fat deposition existing in the dry-cured Bísaro loin lines [22,26] resulted in the rapid release of fluid contained in them, justifying the variation found in the texture parameters of our work. All of them showed a relatively large range (6.56–2.44; 5.44–2.13; 6.11–3.44 for hardness, chewiness and juiciness, respectively) with average values of 3.90, 3.62 and 5.10, respectively, and standard deviations of 1.23, 0.85 and 0.64, respectively. Similar margins were obtained in the study by Revilla et al. [40,49] on “Cecina” dried beef [49]. On the other hand, in a study conducted by Ruivo [50], the author constates that higher body weights have repercussions such as an increase in subcutaneous and intermuscular fat content, an increase in the myoglobin ratio and a decrease in water loss from the muscles, which are in accordance with our results of chemicals, which are not presented here. Therefore, the development of acceptable texture characteristics in these meat products is very important for the successful marketing of high-quality products, as this study aims to achieve. The texture properties include a complex process in which both ingredients and traditional processing steps have a vital role. According to Domínguez et al. [43], the reactions that release characteristic aromas (like hexanals) present a rancid aroma at high amounts, whereas at low amounts, it releases a pleasant grassy aroma. Odor is also an important attribute that interacts with flavor because the ripening degree (proteolysis and lipolysis changes that occur in the drying process) reflects the ripening odor compounds that become a significant part of the meat’s flavor [45,51,52]. Also, Leite et al. [45] found in their work with similar conditions for the production of cured Bísaro loins that specific ingredients in their formulation (for example, garlic) can be detected in this type of product. Therefore, the range values of odor could be partially due to biochemical changes during the drying period, which were very high compared with those of Seong et al. [53].

Regarding appearance, mean values for fat color and color were lower (3.11 and 4.05, respectively) than those described by Revilla et al. [49] and were within those obtained by Seong et al. [53], with a value of 4.75 given by tasters for loins ripened for 60 days in their study. The color of meat is closely associated with freshness, myoglobin content and technological quality traits such as pH or water-holding capacity, which, in Bísaro pork meat, indicate significant variations in color and tenderness [22,26,45]. These indicators play a crucial role in visual attraction and sensory acceptability and are directly related to financial losses for the industry if they do not meet consumer demands [11,13]. Also, in the process of producing, substances such as drying conditions, ripening time, condiments, salt, nitrites and nitrates are added, which directly influence the final color ratios [4,50].

Other attributes evaluated like androsterone and scatol were included in the flavor parameters in this work. Studies have demonstrated that, due to high levels of intrinsic factors such as age, sex and weight at slaughter, the meat from entire males presents undesirable odors, even with a limit of 150 days of age and 100 kg for the slaughter weight [54]. Our samples had weights close to 100 kg and were from castrated animals, so it was expected that these hormones would not have a high expression in our analysis. According to [55], it was found that pigs intended for fresh consumption should be preferably slaughtered between 60 and 120 kg, whereas those primarily destined for processing should be slaughtered between 140 and 180 kg.

Also, Squires and Bonneau [56] mentioned that the fat from entire males contains a higher quantity of polyunsaturated fatty acids, resulting in meat with a lower oxidation resistance, which leads to lower-quality products and processing difficulties. Autochthonous breeds such as the Bísaro are not suited for high yields of lean meat production, but for processed products, they have highly marbled meat that confers excellent flavor due to the variety of the feed they typically consume [3]. Also, according to other authors [43,45], the oxidation process of principal acid compounds of fat is described as a very important compound that contributes a dry-cured aroma to products. Moreover, it is highly rich in fat-soluble vitamins, including vitamins A and E combined with selenium content, which makes this pork meat less susceptible to oxidative rancidity compared to meat from other animals [5,27]. Regarding the presence of fat, the feeding process can influence the duration of the drying process, specifically affecting the intensity and persistence of flavor [57]. Longer periods for drying positively affect the biochemical and microbiological development of flavor. Flavor generation by drying is part of the process and a natural way of producing aromas in meat products. This has become a consumer need in contrast to the direct addition of flavoring to flavorless meat products [58]. In this way, high range values were expected for flavor intensity and flavor persistence (6.44–5.11 and 6.33–4.56, respectively), in accordance with the findings (4.60 flavor intensity) from the study by Seong et al. [53], who used 60 days of ripening time for the sensory characteristics of dry-cured loins. Also, Leite et al. [45] obtained close values for this attribute in dry-cured loins of different lines of Iberian pigs. These results agree with those reported by other authors [14,15,43], in which juiciness, flavor, the absence of off-flavors, appearance and others were the most important attributes comprising the sensory experience during meat consumption and influencing the final consumer choice. Obtaining meat with the desired characteristics is a challenging objective to achieve [59]. As pork and its derivatives can suffer variations as stated above, quality levels and prices would be directly influenced. Therefore, it is crucial to define the distinctive attributes that characterize these products. Sensory evaluation plays a significant role in this regard, as attributes such as odor, taste, color, texture and even the presence of visual fat are representative of the product. Thus, NIR is a key tool that must be applied to streamline and substantiate the entire process [11].

### 3.2. NIR Analysis

#### 3.2.1. NIR Spectra

The use of support vector machine regression (SVmR) has been shown to be the most suitable for data modeling, after a comparison was made between PLS and SVmR models to evaluate their performance in modeling meat characterization data [22]. In addition, when pre-treatments are applied to reduce and correct possible interferences, the results are improved [11,22,30,36]. In this context, the regression models used to predict the sensory attributes of dry-cured loins were obtained using different kinds of SVR tested (Table 1). Through a hybrid algorithm based on particle swarm optimization (PSO) [39], it was possible to minimize the root-mean-square error (RMSECV) (used as the predictive evaluation criteria) obtained via five-fold cross-validation for the parameters.

In this context, the dry-cured loins’ spectra obtained after each pre-treatment are presented in Figure 1. The NIR spectra of the samples present typical broader peaks related to major constituents of proteins, lipids and water [20,60,61]. The major peaks are assigned in Figure 1c; C-H bonds correspond to 1200 nm, 1710–1760 nm and 2300–2350 nm (first overtone, second overtone and combination band, respectively). The N-H and O-H bonds correspond to the 1450 and 1900 nm regions (second and first overtones, respectively).

#### 3.2.2. NIR Models

The SVR models were calibrated for all sensory attributes using the possible parameter combinations in Table 1. The best results were achieved by applying MSC pre-treatment, MinMax normalization, ε-SVR and the radial base kernel. Table 3 presents the best optimized ε-SVR for each sensory attribute.

For the ε-SVR with the radial base kernel, parameters C, ε and γ were optimized (Table 3) using the PSO algorithm to minimize the RMSECV for five-fold cross-validation. The values of these parameters control the complexity of the regression model and, consequently, the prediction capabilities for new data sets. Low values of ε bring the models closer to the calibration data. However, this excessive adjustment may result in a loss of generalization to predict new data. Parameter γ is related to smoothness, and C is related to the complexity of the regression model. The regression model is spikier and more complex for high values of γ and C [62,63]. Good SVR models were obtained for all sensory attributes, with R^2^ values close to 1 (0.9616–0.9955) and low values for the RMSE (0.0400–0.1031) for the prediction set. Furthermore, the relative standard deviation (RSD) for the prediction set was less than 5% for all sensory attributes, and the confidence interval (95%) contains the ideal point (unit slope and zero intercept) [64] for all models, as shown in Table 3. Figure 2 confirms the good prediction achieved by the ε-SVR models for the sensory attributes. These values are better than those found in the work by Hernández-Ramos et al. [65], who found score variations of R^2^ = 0.51 for the pig aroma parameter, R^2^ = 0.67 for the hardness parameter, R^2^ = 0.78 for the fat color attribute and R^2^ = 0.82 for flavor intensity.

The results for the PLS models for the sensory attributes are shown in Appendix A. The R^2^ values reported in the current study for juiciness (R^2^ = 0.62) disagree with those in the works by Ripoll et al. [66], Prieto et al. [67] and Wang et al. [48], who reported less accurate predictions for juiciness (R^2^ = 0.53; 0.21; 0.17, respectively). In addition, for the flavor intensity parameter, the value found in this study (R^2^ = 0.41) is less accurate but still disagrees with the study by Prieto et al. [67], who achieved a higher value (R^2^ = 0.59).

In addition, for the chewiness parameter, the study by Wang et al. [48] only reached a model with R^2^ = 0.09, whereas in the present study, an R^2^ value of 0.58 was obtained. Also, in the work by Rødbotte et al. [68], obtaining any predictive model for sensory juiciness scores was impossible even after applying mathematical pre-treatment to the spectral data (multiplicative scatter correction). All authors mentioned above stress that predicting textural parameter scores is a complex subject, and attributes are difficult to accurately predict consistently.

All PLS models had poor prediction capabilities, and the Durbin–Watson statistical test (DW) had non-significant probabilities (pDW > 0.05), indicating a lack of a correlation between PLS residuals and a lack of non-linearities in the multivariate signal [64]. However, applying the non-linear SVR model greatly improved the prediction capability of the meat sensory attributes using the NIR spectra because of several key advantages: its ability to capture intricate data patterns that PLS, due to its linear nature, might overlook; its diminished sensitivity to multicollinearity when compared to PLS; and its more effective handling of outliers through a margin-based approach.

## 4. Conclusions

This work shows the potential of NIR as a fast and accurate tool for assessing the sensory attributes (like odor, odor androsterone, odor scatol, lean color, fat color, hardness, juiciness, chewiness, flavor intensity and flavor persistence) of dry-cured Bísaro pork. In this way, this work generated acceptable predictive models of sensory parameters using advanced chemometric techniques. The NIR spectra exhibited characteristic peaks of physico-chemical characteristics related to major components such as proteins, lipids and water. Overall, the present study shows that NIR has potential as an analytical tool for real-time meat quality control. Therefore, this study demonstrates that non-linear SVR models, particularly when applied to NIR spectra, significantly improve the prediction of sensory attributes in dry-cured Bísaro loins by offering a promising method for classifying individual animals in breeding programs (extensive production system) and applying this technique in situ at an industrial level to obtain product recognition characteristics of the Bísaro breed. This research’s potential applications include refining quality assessment methodologies, guiding product development strategies and fostering innovation in meat processing technologies. Implementing precise sensory attribute predictions can elevate product standards, meet consumer preferences and drive advancements in the dry-cured loins industry.

## Figures and Tables

**Figure 1 foods-12-04335-f001:**
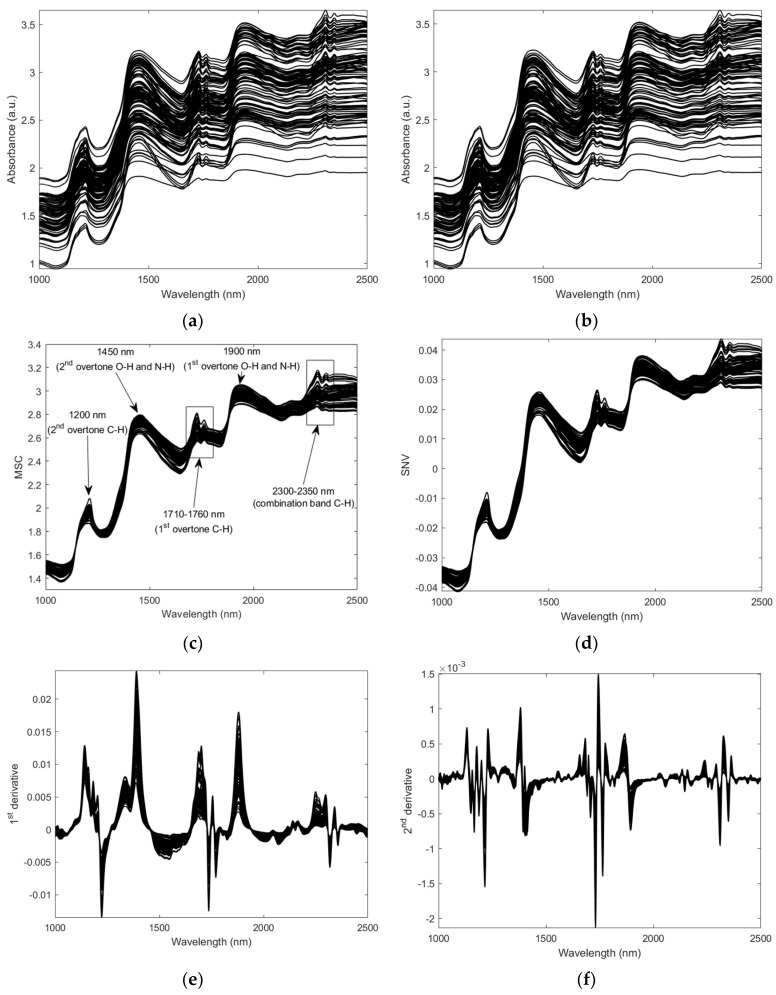
NIR spectra: raw spectra (**a**), smoothed spectra (**b**), MSC (**c**), SNV (**d**), first derivative (**e**) and second derivative (**f**).

**Figure 2 foods-12-04335-f002:**
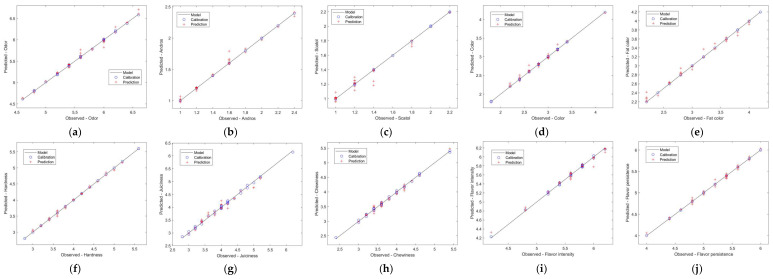
Predicted values for the best ε-SVR models and the observed values for each sensory attribute in the subsigures (**a**) “Odor”, (**b**) “Andros” (**c**) “Scatol”, (**d**) “Color”, (**e**) “Fat color”, (**f**) “Hardness”, (**g**) “Juiciness”, (**h**) “Chewiness”, (**i**) “Favor intensity”, (**j**) ”Flavor persistence”.

**Table 2 foods-12-04335-t002:** Sensory attributes of the forty dry-cured Bísaro loins with min, max and mean represented (*n* = 40).

Attributes	Definition	Min	Max	Mean (±Sd)
Odor	The presence of a typical odor of a dry-cured product [16]	5.13	6.78	5.92 (±0.38)
Andros	The presence of the metabolites of testosterone [45]	1.11	2.38	1.50 (±0.28)
Scatol	The presence of organic compounds that contribute to a fecal odor [45]	1.00	1.78	1.24 (±0.19)
Lean color	The color of the part of the muscle sample [45]	2.88	6.11	4.05 (±0.73)
Fat color	The color intensity and brightness of the fat [45]	1.56	4.89	3.11 (±0.85)
Hardness	The force necessary to penetrate the meat with incisors [16]	2.44	6.56	3.90 (±1.23)
Juiciness	The amount of juice given off by the sample when chewed [16]	3.44	6.11	5.10 (±0.64)
Chewiness	The number of times the sample must be chewed before it can be swallowed [16]	2.13	5.44	3.62 (±0.85)
Flavor intensity	The intensity of the overall flavor of the samples [16]	5.11	6.44	5.89 (±0.33)
Flavor persistence	The persistence of the overall flavor of the mouthfeel [16]	4.56	6.33	5.67 (±0.42)

Sd—standard deviation; Andros—odor androsterone; Min—minimum; Max—maximum; point scale with the extremes representing either the minimum (low-intensity sensation) or the maximum (high-intensity sensation).

**Table 3 foods-12-04335-t003:** Parameters and figures of merit of the best ε-SVR models obtained for each sensory attribute.

				Calibration *		Prediction *		
Attribute	C	ε	γ	RMSE	R^2^	RMSE	R^2^	RSD (%)
Odor	23.43	0.0161	0.0227	0.0155	0.9995	0.0549	0.9888	0.98
Andros	18.11	0.0010	0.0258	0.0011	1.0000	0.0400	0.9892	2.87
Scatol	87.79	0.0051	0.0221	0.0051	0.9998	0.0548	0.9616	4.47
Lean color	39.03	0.0074	0.0151	0.0072	0.9998	0.0507	0.9853	1.85
Fat color	100.0	0.0010	0.0446	0.0010	1.0000	0.0685	0.9878	2.26
Hardness	100.0	0.0010	0.0257	0.0011	1.0000	0.0403	0.9955	1.03
Juiciness	34.28	0.0522	0.0155	0.0499	0.9966	0.1031	0.9705	2.65
Chewiness	40.46	0.0395	0.0090	0.0374	0.9966	0.0674	0.9800	1.81
Flavor intensity	53.00	0.0252	0.0135	0.0240	0.9974	0.0554	0.9876	0.98
Flavor persistence	51.00	0.0010	0.0124	0.0011	1.0000	0.0417	0.9907	0.80

* All models used MSC pre-treatment, MinMax normalization and radial base kernel; MSC—multiplicative scatter correction; Andros—odor androsterone; C/γ/ε PSO—particle swarm optimization parameters; RSD—percentual relative standard deviation; RMSE—root-mean-square error; R^2^—coefficient of determination.

## Data Availability

Data are contained within the article.

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
