# Peer review of "Can Near-Infrared Spectroscopy Replace a Panel of Tasters in Sensory Analysis of Dry-Cured Bísaro Loin?"

_foods, 2023, doi:10.3390/foods12234335_

Round 1

Reviewer 1 Report

Comments and Suggestions for Authors

The manuscript foods-2725265 entitled "Can NIR replace a panel of tasters in sensory analysis of dry-cured Bísaro loin?

The manuscript is well-written, and informations and the contents of the manuscript are of great value. However, there are some areas that require further improvement to enhance the overall quality of the paper

1)     Please write the full form of NIR in the main title of the manuscript. In main title, we should avoid abbreviations to increase visibility to larger audience

2)     Keywords: Please replace those keywords with other suitable words which already appeared in the main title of the manuscript such as dry-cured Bísaro loin, NIR analysis, and sensory attributes. No need to repeat these words again in keywords

3)     The introduction section does not explain what was done on topic analyzed. Please refer some already published data on the relevant topic. For instance, the main objective of this study is comparison of sensory analysis with NIR technique but the authors too much talked about the irrelevant topics such as animal breeds etc. I will recommend to discuss the advantages and disadvantages of both techniques in detail.

4)      Results and discussion: I have read carefully the results and discussion section and I found that the appropriate discussion of the results obtained. However, as occurs with the Introduction this section is very disorganized, without any connection between paragraphs with irrelevant comments

5)     Conclusion: This section is not adequate rewrite please.

Author Response

Dear review,

All modifications were made following the reviewer`s suggestions and comments, and

responses to their comments are also attached. Thanks to their recommendations,

significant modifications were made throughout the manuscript.

Thank you for your attention.

Answers to Reviewer 1

Comments and Suggestions for Authors

The manuscript foods-2725265 entitled "Can NIR replace a panel of tasters in sensory analysis of dry-cured Bísaro loin?”

The manuscript is well-written, and informations and the contents of the manuscript are of great value. However, there are some areas that require further improvement to enhance the overall quality of the paper

Response: We thank the reviewer for the attention given to reviewing the article and for the comments, which we took advantage of to improve some areas the manuscript.

1)     Please write the full form of NIR in the main title of the manuscript. In main title, we should avoid abbreviations to increase visibility to larger audience

Response: Suggestion accepted; changes have been made in the revised version of the

manuscript.

2)     Keywords: Please replace those keywords with other suitable words which already appeared in the main title of the manuscript such as dry-cured Bísaro loin, NIR analysis, and sensory attributes. No need to repeat these words again in keywords

Response: Suggestion accepted; changes have been made in the revised version of the

manuscript.

3)     The introduction section does not explain what was done on topic analyzed. Please refer some already published data on the relevant topic. For instance, the main objective of this study is comparison of sensory analysis with NIR technique but the authors too much talked about the irrelevant topics such as animal breeds etc. I will recommend to discuss the advantages and disadvantages of both techniques in detail.

Response: We thank you for your suggestion; some changes have been made in the revised version of the manuscript. However, we do not agree that the animal breed approach is an “irrelevant topic”, since even if the focus of the work is an approach to NIR analysis in estimating the physicochemical composition of the dry-cured product under analysis aligned with sensory evaluation, we can obtain a sensory characterization of it for a specific type of market. To this end, it is essential to study a breed such as Bísaro, where there are already premium products with quality labels, in order to offer consumers, the option of another product with excellent characteristics, such as cured loin, which can also be awarded these labels.

4)      Results and discussion: I have read carefully the results and discussion section and I found that the appropriate discussion of the results obtained. However, as occurs with the Introduction this section is very disorganized, without any connection between paragraphs with irrelevant comments

Response: Suggestion accepted; changes have been made in the revised version of the

manuscript.

5)     Conclusion: This section is not adequate rewrite please.

Response: Suggestion accepted; changes have been made in the revised version of the

manuscript.

Reviewer 2 Report

Comments and Suggestions for Authors

Here are some points of improvement:

The abstract could benefit from a more detailed and structured explanation of the methodology. For instance, you can briefly explain the steps involved in the NIR analysis, the preprocessing of spectra, and how the SVR models were developed. It's essential to define acronyms upon first use. For instance, NIR, SVR, PLS, and PSO should be defined the first time they are mentioned to ensure clarity for readers who may not be familiar with these terms. The abstract mentions different data preprocessing techniques and SVR models, but it could be improved by briefly explaining why these techniques were chosen and how they are expected to improve the precision of sensory evaluations. The abstract mentions that the predictive models exhibited acceptable results, but it does not provide specific details on these results. Including key findings, such as R-squared values and RMSE values, can give readers a better understanding of the study's outcomes. While the abstract concludes that the research highlights the potential of advanced analytical techniques, you can further emphasize the practical implications or real-world applications of the research findings. Why do these findings matter, and what impact can they have in the field of food quality assessment? Consider including more specific keywords relevant to the study to improve the discoverability of your research. The Results and Discussion section could benefit from improved clarity and organization. Consider breaking it down into subsections to make it easier for readers to follow and find specific information. For example, you could create subsections for sensory data, NIR analysis, model parameters, and results, making it easier to navigate the content. Start the section by providing a brief introduction to the sensory attributes you are analyzing. This would help readers understand the context before diving into the data. In the section discussing sensory attributes (e.g., odor, androsterone, scatol, color, etc.), provide a bit more background information and context for each attribute. Explain why these attributes are important and how they relate to the quality of dry-cured loins. This will help readers who may not be familiar with these attributes. When presenting the sensory data in Table 2, it would be helpful to provide a brief description of each attribute and its relevance. Consider adding a column in the table with a brief explanation for each attribute. After presenting the sensory data, offer a more in-depth interpretation of the results. Discuss the implications of the findings for each attribute. For example, explain how the wide variation in certain attributes may impact the quality of the dry-cured loins and how this relates to consumer preferences or industry standards. Consider incorporating visual aids, such as plots or graphs, to help visualize the sensory data. Visual representations can make it easier for readers to grasp the variations in sensory attributes. When discussing the SVR models, provide a more detailed explanation of the model parameters (C, ε, γ) and their significance. Explain how these parameters affect the model's performance and why specific values were chosen. This would provide more insight into the modeling process. Compare the results obtained from the SVR models with those from the PLS models. Highlight the advantages of using SVR over PLS for predicting sensory attributes and discuss the limitations of PLS models in this context. Conclude the section by summarizing the key findings and their significance. Discuss the implications of the accurate prediction of sensory attributes for the dry-cured loins industry and the potential applications of this research. Ensure that you properly cite previous works and studies mentioned in the text, as this will add credibility to your research.

Comments on the Quality of English Language

Moderate editing of English language required

Author Response

Answers to reviewer 2:

Comments and Suggestions for Authors

Here are some points of improvement:

The abstract could benefit from a more detailed and structured explanation of the methodology. For instance, you can briefly explain the steps involved in the NIR analysis, the preprocessing of spectra, and how the SVR models were developed.

Response: In abstract, a new phrase was introduced addressing this issue:

“This process involved partitioning the data into calibration (67%) and prediction (33%) subsets, using the SPXY algorithm. The model parameters were optimized via a hybrid algorithm based on particle swarm optimization (PSO), to effectively minimize the root mean square error (RMSECV) derived from 5-fold cross-validation and ensure the attainment of optimal model performance and predictive accuracy.”

It's essential to define acronyms upon first use. For instance, NIR, SVR, PLS, and PSO should be defined the first time they are mentioned to ensure clarity for readers who may not be familiar with these terms.

Response: We systematically reviewed the initial occurrences of concepts and abbreviations in both the abstract and the introduction to rectify any discrepancies or repetitions.

The abstract mentions different data preprocessing techniques and SVR models, but it could be improved by briefly explaining why these techniques were chosen and how they are expected to improve the precision of sensory evaluations.

Response: With the addition of the new sentence in the abstract, the comprehensive process undertaken to derive the model is now elucidated, encompassing the steps executed to ensure optimal model performance and predictive accuracy.

However, concerning the query of “ … why these techniques were chosen… “, a subsequent phrase has been incorporated in the section “2.5 Data Analysis” denoting that these methodologies are standard procedures in the development of NIR calibrations.

“These four signal pretreatment methods are conventional procedures that can be tested in combination with various signal normalization techniques to further enhance the optimization of NIR calibration.”

The abstract mentions that the predictive models exhibited acceptable results, but it does not provide specific details on these results. Including key findings, such as R-squared values and RMSE values, can give readers a better understanding of the study’s outcomes.

Response: The abstract presents the overall results about the obtained predictive models:

“The predictive models exhibited acceptable results, characterized by R-squared values close to 1 (0.9616 - 0.9955) and low RMSE values (0.0400 – 0.1031). The relative standard deviation (RSD) for the prediction set remained under 5%.”

While the abstract concludes that the research highlights the potential of advanced analytical techniques, you can further emphasize the practical implications or real-world applications of the research findings. Why do these findings matter, and what impact can they have in the field of food quality assessment?

Response: To the phrase in the abstract:

“This research underscores the potential of advanced analytical techniques to improve the precision of sensory evaluations in food quality assessment.”

a new phrase was added:

“Such advancements have the potential to benefit both the research community and the meat industry by closely aligning their practices with consumer preferences and expectations.”

Consider including more specific keywords relevant to the study to improve the discoverability of your research.

Response: A new keywords were introduced:

food quality assessment; consumers; meat products; Bísaro breed

The Results and Discussion section could benefit from improved clarity and organization. Consider breaking it down into subsections to make it easier for readers to follow and find specific information. For example, you could create subsections for sensory data, NIR analysis, model parameters, and results, making it easier to navigate the content.

Response: We thankful the suggestions; We tried to subsection the information in this topic to be clearer and more organized.

Added “3.1.1 Sensory attributes”; “3.1.2 Sensory data”; “3.2.1 NIR spectra”; 3.2.2 “NIR models”

Start the section by providing a brief introduction to the sensory attributes you are analyzing. This would help readers understand the context before diving into the data.

Response: In subsection 3.1.1, the following text “For decades, DA has been used for the characterization and discrimination of meat and meat products. For that, the descriptor chosen are vital in these analyses because all specific attributes allow a complete characterization of the dry-cured product in study and describe it in more detail. Also, it was applied with a 9-point hedonic scale (used for the assessment of the linking of the product) because this hedonic test could provide in-formation about different products, textures, composition, etc., which would help to better understand the taster’s answer. Therefore, in dry-cured meat products the main sensory descriptors are basically grouped in visual appearance, texture, and flavor. Among all appearance characteristics, visual meat color (fat and muscle) was the major criterion used to choose meat appearance. With relation to texture attribute, is one of the most important eating quality at-tributes which could be defined as the overall impression related to texture properties. In SE of meat and meat products are including juiciness (defined as sensation of the water content of the sample and the saliva released with the stimulation of the salivary glands), hardness and chewiness (attributes based on the interaction between the textural characteristics and “mouthfeel”). Also, the flavor attribute will often be perceived when the combination of odor (prior to consumption and retronasal olfaction) and taste perception, along with trigeminal perception, leads to flavor perception. In addition, two odors likely to be detected in animals with a high live carcass weight and/or poorly castrated were analyzed such as androsterone and skatole hormones of animal origin).” has been inserted in order to respond to what was requested.

We believe that after reading the introduction and with this new information (that completes the previously one), readers will better understand the context before diving into the data.

In the section discussing sensory attributes (e.g., odor, androsterone, scatol, color, etc.), provide a bit more background information and context for each attribute. Explain why these attributes are important and how they relate to the quality of dry-cured loins. This will help readers who may not be familiar with these attributes. Discuss the implications of the findings for each attribute. For example, explain how the wide variation in certain attributes may impact the quality of the dry-cured loins and how this relates to consumer preferences or industry standards.

Response: Throughout the text, the discussion section shows the context of the attribute and its importance in the quality of the cured loins analyzed. However, changes have been made in order to help readers understand the context.

A new phrases were added:

“In a general way, it can be concluded that the data obtained from the evaluation of trained tasters give quantitative and objective information despite the wide range. In fact, with the AD test (hedonic evaluation) used, it was possible to obtain information on the magnitude of the liking or disliking of these sensory attributes studied. Several publications describe multiple attributes evaluated during the sensory analysis of meat products.”

“Therefore, the development of acceptable texture characteristics in this meat products is very important for the successful marketing of high-quality products, as this study aims to achieve. The texture properties include a complex process in which both ingredients and traditional processing steps have a vital role. According to Domínguez et al. the reactions that release the characteristic aromas (like hexanals) present a rancid aroma at high amounts, while at low amounts, it gives a pleasant grassy aroma. The odor is also important attribute inter with flavor, because the ripening degree (proteolysis and lipolysis changes that occur in drying process) reflects the ripening odor compounds that become a significant part of meat flavor. Also, the authors Leite et al.  found in their work with similar conditions of production of cured Bísaro loins that specific ingredients in their formulation (for ex. garlic) can be detected in this type of product. Therefore, the range values of odor could be partially due to the biochemical changes during the drying period, very high compared with Seong et al. work.”

“As well Leite et al. obtained close values this attribute in dry-cured loins of different lines of Iberian pigs. These results agree with those reported by other authors in which juiciness, flavor, absence of off-flavors, appearance and others were the most important attributes comprising the sensory experience during meat consumption and in-fluence the final consumer choice. Obtaining meat with the desired characteristics is a challenging objective to achieve.”

When presenting the sensory data in Table 2, it would be helpful to provide a brief description of each attribute and its relevance. Consider adding a column in the table with a brief explanation for each attribute. After presenting the sensory data, offer a more in-depth interpretation of the results.

Response: Suggestion accepted; The column was added to Table 2 with a brief description of each attribute and the results the results are explained below.

Consider incorporating visual aids, such as plots or graphs, to help visualize the sensory data. Visual representations can make it easier for readers to grasp the variations in sensory attributes.

Response: We appreciate the suggestion of incorporating visual aids, such as plots or graphs, to help visualize the sensory data. However, we believe that we should not introduce these graphs or plots since we want to focus our work on a Nir analysis approach in a complementary way and innovate other existing works that have a more traditional approach.

When discussing the SVR models, provide a more detailed explanation of the model parameters (C, ε, γ) and their significance. Explain how these parameters affect the model's performance and why specific values were chosen. This would provide more insight into the modeling process. Compare the results obtained from the SVR models with those from the PLS models.

Response: In section “3.2 NIR analysis” a description of these model’s parameters were introduced:

“For the ε-SVR with the radial base kernel, the parameters C, ε, and γ were optimized (Table 3) using the PSO algorithm to minimize the RMSECV for 5-fold cross-validation. The values of these parameters control the complexity of the regression model and, consequently, the prediction capabilities for new data sets. Low values of ε bring the models closer to the calibration data. However, this excessive adjustment may conduct in a loss of generalization to predict new data. The parameter γ is related to smoothness and C to the complexity of the regression model. The regression model is spikier and more complex for high values of γ and C.”

Highlight the advantages of using SVR over PLS for predicting sensory attributes and discuss the limitations of PLS models in this context.

Response: The last phrase of chapter 3:

“However, applying a non-linear SVR model greatly improved the prediction capability of the meat sensory attributes using the NIR-spectra-”

was changed to:

“However, applying a non-linear SVR model greatly improved the prediction capability of the meat sensory attributes using the NIR-spectra, because of several key advantages: its ability to capture intricate data patterns that PLS, due to its linear nature, might overlook; its diminished sensitivity to multicollinearity when compared to PLS; and its more effective handling of outliers through a margin-based approach.”

Conclude the section by summarizing the key findings and their significance. Discuss the implications of the accurate prediction of sensory attributes for the dry-cured loins industry and the potential applications of this research.

Response: The phrase in the conclusion section

“This study involved the assessment of various sensory attributes (like odor, androsterone, scatol, color, fat color, hardness, juiciness, chewiness, flavor intensity, and flavor persistence) of dry-cured loins using NIR spectra and advanced chemometric techniques.” was rewriting for new one: “This work shows the potential of NIR as a fast and accurate tool for assessing the sensory attributes (like odor, odor androsterone, odor skatole, lean color, fat color, hard-ness, juiciness, chewiness, flavor intensity, and flavor persistence) of dry-cured Bísaro pork. In this way, this work generated acceptable predictive models of the sensory parameters using advanced chemometric techniques.”

The phrase “Overall, the study demonstrated that non-linear SVR models, particularly when applied to NIR spectra, significantly improved the prediction of sensory attributes in dry-cured loins.” Was rewriting for new one: “Therefore, the study demonstrated that non-linear SVR models, particularly when applied to NIR spectra, significantly improved the prediction of sensory attributes in dry-cured Bísaro loins by offering a promising method for classifying individual animals in breeding programs (extensive production system) and applying this technique in situ at an industrial level to obtain product recognition characteristics of the Bísaro breed.”

The phrase “This highlights the potential of advanced analytical techniques to enhance the accuracy of sensory evaluation in food quality assessment.” Was removed and a new phrase was introduced: “This research's potential applications include refining quality assessment methodologies, guiding product development strategies, and fostering innovation in meat processing technologies. Implementing precise sensory attribute prediction can elevate product standards, meet consumer preferences, and drive advancements in the dry-cured loins industry.”

Ensure that you properly cite previous works and studies mentioned in the text, as this will add credibility to your research.

Response: Can the reviewer indicate in which section there is a lack of discussion of results with other authors' work?

With regard to this study, we have made an effort to discuss results with other sensory analysis studies, as can be seen in section 3.2 NIR analysis.

Comments on the Quality of English Language

Moderate editing of English language required

Response: An English revision was carried out to improve the English language.

Reviewer 3 Report

Comments and Suggestions for Authors

General comments

This article is one of the few studies on the possibility of using objective methods of assessing the sensory characteristics of meat products instead of traditional subjective methods of consumer evaluation. Although the sensory evaluation of dry-cured Bisaro loin was performed by nine trained tasters, the evaluation scale used is not precisely described, and the number of grades (ratings) is relatively large, which reduces the accuracy of the evaluation. The use of the NIR technique seems to be a good solution to increase the accuracy of the assessment, although it is man and his senses that are the recipient of the sensory quality of a given product and his decision to choose it. Overall, the study demonstrated that non-linear SVR models, particulary when applied to NIR spectra significantly improved the prediction of sensory attributes in dry-cured loins.

Specific comments

L35 like pig meat aroma?

L43 nutritional content or nutritional value?

L139 How old were the pigs, what BW

L141 What slaughter age?

L142 Maybe write something about nutrition. The composition of the diet affects the chemical composition of meat and its sensory characteristics.

L165 What was the product yield (in percent)? Relative to initial weight?

L165 How often were the environmental parameters where the loins were dried?

L170 What was the age and gender of the testers?

L204 What was the weight of the tested sample?

L206 4,000 to 10,000 cm-1 sectal range?

In Table 2 Maybe „Odor androsterone” – it will fit

Under the table no 2 Explanations, how many degrees the scale had, what was the lowest and highest score

Editorial requirements

L428-429 Name of the conference according to the instructions for the author

Long dash with insert the symbol "" instead of the short "-" from the keyboard

Dot after each part of abbreviation name journal

Is L502, L507, L536 an abbreviation name journal?

L551 Is "LWT" the full name of the journal?

Author Response

Dear review,

All modifications were made following the reviewer`s suggestions and comments, and

responses to their comments are also attached. Thanks to their recommendations,

significant modifications were made throughout the manuscript.

Thank you for your attention.

Answers to reviewer 3

Comments and Suggestions for Authors

General comments

This article is one of the few studies on the possibility of using objective methods of assessing the sensory characteristics of meat products instead of traditional subjective methods of consumer evaluation. Although the sensory evaluation of dry-cured Bisaro loin was performed by nine trained tasters, the evaluation scale used is not precisely described, and the number of grades (ratings) is relatively large, which reduces the accuracy of the evaluation. The use of the NIR technique seems to be a good solution to increase the accuracy of the assessment, although it is man and his senses that are the recipient of the sensory quality of a given product and his decision to choose it. Overall, the study demonstrated that non-linear SVR models, particulary when applied to NIR spectra significantly improved the prediction of sensory attributes in dry-cured loins.

Response: We thank the reviewer for the attention given to reviewing the article and for the comments, which we took advantage of to improve the manuscript.

Specific comments

L35 like pig meat aroma?

Response: Suggestion accepted; changes have been made in the revised version of the

manuscript.

L43 nutritional content or nutritional value?

Response: Suggestion accepted; changes have been made in the revised version of the

manuscript.

L139 How old were the pigs, what BW

Response: Suggestion accepted; changes have been made in the revised version of the

manuscript.

L141 What slaughter age?

Response: Suggestion accepted; changes have been made in the revised version of the

manuscript.

L142 Maybe write something about nutrition. The composition of the diet affects the chemical composition of meat and its sensory characteristics.

Response: We thank you for your suggestion; We agree that composition of the diet can affects the characteristics of the meat; However, this study used animals from two farms in the region where the work is carried out, so the animal feed administered is in accordance with a basic diet (we don't have access to the specific nutritional components of the diet) typical of these indigenous breeds determined by the production industry (Bísaro industry – Salsicharia Tradicional, Lda, ®, Portugal).  We note the change in the revised version.

L165 What was the product yield (in percent)? Relative to initial weight?

Response: This was not mentioned because it is an internal issue inherent to the curing process stipulated by the meat manufacturing industry (Bísaro industry – Salsicharia Tradicional, Lda, ®, Portugal). Throughout the drying process until the end of the drying process, the dry-cured loin samples have a significant loss of between 30 and 40 % of their initial weight. The authors do not consider this to be relevant data to mention in the study.

L165 How often were the environmental parameters where the loins were dried?

Response: The meat manufacturing industry (Bísaro industry – Salsicharia Tradicional, Lda, ®, Portugal) where the product curing process takes place is a company specializing in the production of animal products from Bísaro breed. This company follows strict quality control standards (IFS) where all the products are made according to adjusted manufacturing flowcharts that respect times and temperatures that are controlled on a daily basis by the responsible technical staff who work there.

L170 What was the age and gender of the testers?

Response: We don't consider it relevant to mention this information in the paper. The tasters' ages range from 20 to 60 years old, and 7 was female and 2 was male.

L204 What was the weight of the tested sample?

Response: We don't consider it relevant to mention this information in the paper. The petri dishes with a diameter of approximately 9 cm used in this study have a capacity of around 200g of sample. No specific weight was set since the NIR instrument can analyze the sample from 100g.

L206 4,000 to 10,000 cm-1 sectal range?

Response: The suggestion has been verified. The information "spectral range" as mentioned in the manuscript is in the correct form.

In Table 2 Maybe „Odor androsterone” – it will fit

Response: Thank you for your suggestion. However, making this change means reducing the size of the table, which could be detrimental to the reader.

Under the table no 2 Explanations, how many degrees the scale had, what was the lowest and highest score

Response: The information was added in the revised version of the  manuscript.

Editorial requirements

L428-429 Name of the conference according to the instructions for the author

Response: Suggestion accepted; changes have been made in the revised version of the

manuscript.

Long dash with insert the symbol "" instead of the short "-" from the keyboard

Response: Suggestion accepted; changes have been made in the revised version of the

manuscript.

Dot after each part of abbreviation name journal

Response: Suggestion accepted; changes have been made in the revised version of the

manuscript.

Is L502, L507, L536 an abbreviation name journal?

Response: Suggestion accepted; changes have been made in the revised version of the

manuscript.

L551 Is "LWT" the full name of the journal?

Response: No it isn`t. Is the abbreviation. The full name of the journal is LWT-Food science and technology

Round 2

Reviewer 2 Report

Comments and Suggestions for Authors

Accept in present form